



# Fluorescence characteristics, absorption properties, and radiative effects of water-soluble organic carbon in seasonal snow across northeastern China

Xiaoying Niu[1], Wei Pu[1], Pingqing Fu[2], Yang Chen[1], Yuxuan Xing[1], Dongyou Wu[1], Ziqi Chen[1], Tenglong Shi[1], Yue Zhou[1], Hui Wen[1], Xin Wang[1,2]

[1]Key Laboratory for Semi-Arid Climate Change of the Ministry of Education, College of Atmospheric Sciences, Lanzhou University, Lanzhou 730000, China

[2]Institute of Surface-Earth System Science, Tianjin University, Tianjin 300072, China

*Correspondence to*: Xin Wang (wxin@lzu.edu.cn)

**Abstract.** Although water-soluble organic carbon (WSOC) in the cryosphere can significantly influence the global carbon cycle and radiation budget, WSOC in the snowpack has received little scientific attention to date. This study reports the fluorescence characteristics, absorption properties, and radiative effects of WSOC based on 34 snow samples collected from sites in northeastern China. Sampling sites were divided into five groups, comprising southeastern Inner Mongolia (SEIM), northeastern Inner Mongolia (NEIM), the south of northeastern China (SNC), the north of northeastern China (NNC), and the Changbai Mountain area (CBM). Together, these groups represent a significant degree of regional WSOC variability, with concentrations ranging from $0.50 \pm 0.19$ to $5.70 \pm 3.68$ µg g$^{-1}$ (mean = $3.59 \pm 3.19$ µg g$^{-1}$). We then identified the three principal fluorescent components of WSOC as (1) a high-oxygen humic-like component (HULIS-1) of terrestrial origin, (2) a low-oxygen humic-like component (HULIS-2) of mixed origin, (3) and a protein-like component (PRLIS) derived from autochthonous microbial activity. In SEIM, a region dominated by desert and exposed soils, the WSOC content exhibits the highest humification index (HIX) but the lowest fluorescence (FI) and biological (BIX) indices; the fluorescence signal is mainly attributed to HULIS-1, and thus implicates soil as the primary source. By contrast, the HIX (FI and BIX) value was the lowest (highest) and PRLIS most intense in the remote grasslands and forested areas of NEIM, suggesting a primarily biological source. For SNC and NNC, both of which are characterized by intensive agriculture and industrial activity, the fluorescence signal is dominated by HULIS-2 and the HIX, FI, and BIX values are all moderate, indicating mixed origins for WSOC (anthropogenic activity, microbial activity, and soil). We also observed that, throughout northeastern China, the light absorption of WSOC is dominated by HULIS-1,



followed by HULIS-2 and PRLIS. The contribution of WSOC to albedo reduction (average concentration
3.6 µg g$^{-1}$) in the ultraviolet–visible (UV–vis) band is approximately half that of black carbon (BC:
average concentration 0.6 µg g$^{-1}$); radiative forcing is 3.8 (0.8) W m$^{-2}$ in old (fresh) snow, equating to
19 % (17 %) of the radiative forcing of BC. These results indicate that WSOC has a profound impact on
snow albedo and the solar radiation balance.
**1    Introduction**
Seasonal snow plays a significant role in Earth's solar radiation energy budget owing to its high
reflectivity (Beniston et al., 2017; Usha et al., 2020; Xie et al., 2018). In recent decades, however, the
extent of snow-covered areas has trended downward, partially as a result of the presence of light-
absorbing particles (LAPs) in the snowpack (Barnett et al., 2008; Groisman et al., 1994; Dumont et al.,
2014; Keegan et al., 2014). Black carbon (BC), organic carbon (OC), mineral dust (MD), and biota
comprise the principal light-absorbing particles in seasonal snow (Di Mauro, 2020; Qian et al., 2015),
which together serve to lower surface albedo and impose a positive radiative forcing (Dumont et al.,
2014; Hansen and Nazarenko, 2004; Warren and Wiscombe, 1980; Zhang et al., 2017). Concurrently,
LAPs absorb solar radiation, thereby accelerating snow melt (Li et al., 2021b). Ultimately, the disruption
of the global radiative balance due to LAPs has important implications for regional and global climate
change (Skiles et al., 2018).
Snowpack BC and MD have been the focus of considerable research in snow-covered regions worldwide
(Li et al., 2021a; Zhang et al., 2018; Antony et al., 2014; Hegg et al., 2010; Doherty et al., 2014; Wang
et al., 2015). As the most important LAP (Bond et al., 2013; Doherty et al., 2010; Wang et al., 2014b),
the radiative efficiency of snowpack BC can be more than three times greater than that of carbon dioxide
(Flanner et al., 2007), and MD, another important snowpack LAP, is also known to alter the cryospheric
environment owing to its light-absorbing properties (Di Mauro et al., 2015; Painter et al., 2007; Sarangi
et al., 2020; Shi et al., 2021). Recently, researchers have also begun evaluating the influence of biomes
on global snow albedo (Hotaling et al., 2021). In contrast, however, the role of OC remains poorly
understood because of its complex composition and a relative dearth of OC-focused research.
Consequently, substantial uncertainty surrounds the origins, optical properties, and radiative effects of
snowpack OC.



A recent study has reported that the storage of OC in mountain glaciers and ice caps (~11 % of Earth's
land surface) could be as high as 6 petagrams (Pg; Hood et al., 2015), the majority of which is water-
soluble organic carbon (WSOC). WSOC is one of the largest sources of bioavailable organic carbon in
aquatic ecosystems (Battin et al., 2009). Moreover, as the chief absorber of WSOC, water-soluble brown
carbon (WS-BrC) can absorb significant measures of solar radiation in the ultraviolet–visible (UV–vis)
wavelengths (Murphy et al., 2008). For instance, in their analysis of 21 snow samples from Arctic and
Antarctic, Anastasio and Robles (2007) observed that 50 % of the total light absorption coefficients at
wavelengths > 280 nm might be attributed to organic chromophores of WSOC. In surficial snow samples
from Barrow, Alaska, Beine et al. (2011) reported that WSOC occupies almost the entire absorption
spectrum of the photochemically active region (300–450 nm), and Feng et al. (2016) observed that
absorption in cryoconite samples from the central Tibetan Plateau is dominated by WSOC components
in the 300–350 nm range. Similarly, Yan et al. (2016) measured WSOC in glacial snow from Laohugou,
northern Tibetan Plateau, where they found that the radiative forcing is ~10 % that of BC. Together,
these studies indicate that WSOC plays a key role in global snowpack energy absorption (Niu et al., 2018;
Zhang et al., 2020). Nevertheless, we note that previous research on cryospheric WSOC has focused
largely on alpine glaciers and polar regions; the extensive mid-latitude regions impacted by seasonal
snowpack remain relatively understudied.
The composition of WSOC is typically complex, and characteristics of fluorescence and absorption can
vary widely among the different components. Nonetheless, recent studies have tended to treat WSOC as
a single entity and focus on the overall impacts, such that the specific roles of individual components are
poorly constrained. One commonly used analytical method for distinguishing the components and
properties of fluorescence is the fluorescence excitation-emission matrix (EEM), which has the
advantage of high sensitivity and small sample size (Coble, 1996; Kowalczuk et al., 2005). First applied
in oceanic contexts (Coble et al., 1990), EEM has been gradually extended to lakes, fog water, rainwater,
and atmospheric aerosols in addition to glacial meltwater, ice cores, and snow (Birdwell and Valsaraj,
2010; Huguet et al., 2009; McKnight et al., 2001). Concurrently, parallel factor analysis (PARAFAC) is
an effective approach for extracting from complex EEMs the individual fluorescence components and
their corresponding fluorescence information, thus making EEM–PARAFAC a direct and viable means
for exploring sources of WSOC. For example, Zhou et al., (2019b) used EEM–PARAFAC to identify


the multiple sources of WSOC measured in seasonal snow in northwestern China. Accordingly, we have
applied EEM–PARAFAC in our analysis of snow samples for this study.
Northeastern China supports an extensive snowpack during winter and spring. As a major industrial and
agricultural center, this region is also the principal source of heavy airborne pollutants that are
incorporated into seasonal snow via wet and dry deposition (Wang et al., 2017). Coupled with intensive
tilling of farmland, the geographical proximity of northeastern China to neighboring desert regions also
provides a source of soil organic matter that becomes entrained into the snowpack (Wang et al., 2013b).
Compared with research on BC-snow mixing ratios and their radiative impact in northeastern China
(Dang et al., 2017; Huang et al., 2011; Pu et al., 2019), studies of WSOC are still in their infancy. To
address this deficiency, we analyzed 34 samples of seasonal snow collected in December 2020 and
January 2021 to make the first investigation of the fluorescence characteristics, absorption properties,
and radiative effects of WSOC in northeastern China. Specifically, we applied EEM–PARAFAC to
identify the origins and fluorescence characteristics of snowpack WSOC, after which we derived
individual absorption contributions for each WSOC component using fluorescence data, an absorption
data series, and an attribution method. Finally, we estimated the reduction of snow albedo and radiative
forcing caused by WSOC and BC via the Spectral Albedo Model for Dirty Snow (SAMDS) radiative
transfer model.
**2    Methods**
**2.1    Sample collection**
During the months of January and December 2020 and January 2021, we collected 34 snow samples
from sites across northeastern China, including the eastern part of Inner Mongolia and Heilongjiang and
Jilin provinces. Sample numbers were set following previous campaigns (Pu et al., 2017; Wang et al.,
2013b, 2017), with the exception that samples from the Changbai Mountain area are numbered
individually. The geographical distribution of sampling sites and respective land-cover types are shown
in Figure 1a; our sites are characterized by five land-cover types, including forest, grassland, desert,
cropland, and frozen lake/river (Fig. 1b–g). On the basis of these geographical and environmental
classifications, we divided the sampling sites into five broad regions: southeastern Inner Mongolia (SEIM;
Q494–495, Q497–499), the south of northeast China (SNC; Q470–471, Q473–474, Q477, Q484, Q486–





Q489, Q491–Q493, Q501), the north of   northeast China (NNC; Q480–483), the Changbai Mountain
area (CBM; CM1–CM2, CM5, CM11, CM13–CM14), and northeastern Inner Mongolia (NEIM; Q440,
Q443, Q447, Q449, Q454).

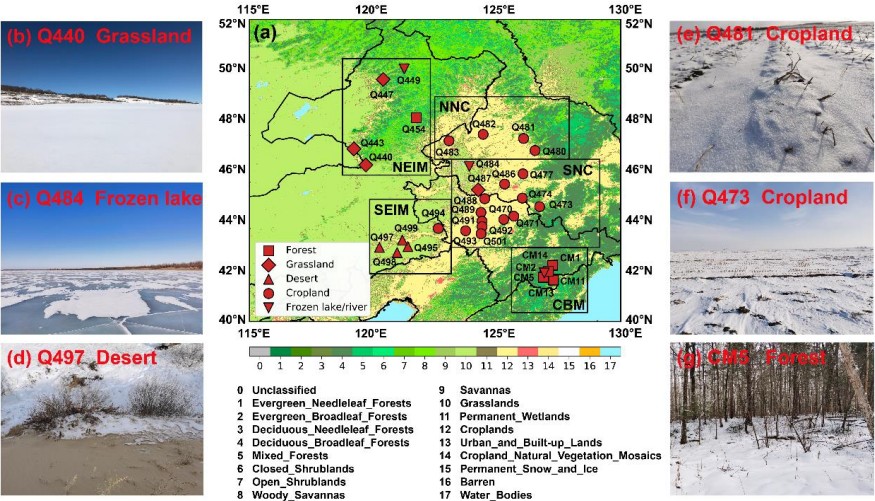

**Figure1: (a) Information on sampling site distributions in northeastern China, including the land cover type,**
**site number, and grouping. Land cover types are derived from Collection 5.1 of the MODIS global land cover**
**type dataset (MCD12Q1: https://lpdaac.usgs.gov/products/mcd12q1v006/) and are indicated by specific**
**colors and symbols relative to sampling sites. Sampling sites are divided into the five groups defined by black**
**rectangles. (b–g) Photographs depicting the typical snow and ground-cover conditions of our various sampling**
**sites.**
Our sample sites were chosen at random but had to be located ≥ 20 km from cities and villages and at
least a kilometer upwind of roads or railroads to minimize the influence of single-point pollution sources
and to ensure the broadest regional representation. Furthermore, we performed sample collection oriented
toward the wind to avoid contamination from personnel. At each site, we used a sterile disposable shovel
to collect 0–5 cm-thick samples of surface snow, which were subsequently stored in sterile Whirlpak
(Nasco, WI, USA) bags. For snow depths < 5 cm, we determined the sampling depth according to the
actual conditions to avoid inducting significant soil impurities during sampling. Snow samples were
melted at room temperature (25 ℃) and stored in a freezer at −20 ℃ until analysis in the laboratory. For
more operational details, we refer the reader to Wang et al. (2013b).



## 2.2 Chemical species analysis

After collection, samples were melted at room temperature (25 °C) before being filtered using a disposable sterile syringe (Jiangnan, Anhui, China) and 0.45 μm pore-sized polytetrafluoroethylene (PTFE) filter (Jingteng, Tianjin, China) (Chen et al., 2019a). Prior to analysis, filtrates were stored in pre-baked (4 hours at 450 °C) glass vials in a freezer. For each sample, we used a total organic carbon analyzer (Aurora 1030W, OI Analytical, TX, USA) to measure the concentration of WSOC; measurement detection limits and relative standard deviations were 2 μg L$^{-1}$ and 1 %, respectively. Blank-corrected concentrations are provided in Table S1.

We used 0.4 μm pore-sized polycarbonate filter membranes (Whatman, USA) to isolate BC and other insoluble particles, following the protocols outlined by Shi et al. (2020) and Wang et al. (2014b), after which we employed a custom-developed two-sphere integrating-sandwich (TSI) filter-based spectrophotometer to measure particle absorption. Coupled with the mass of filtered meltwater, these optical measurements were then converted to snowpack BC concentrations. To make these calculations, we applied a BC mass-absorption coefficient (MAC) and absorption Ångström exponent (AAE) of 6.3 m$^2$ g$^{-1}$ (550 nm) and 1.1, respectively, after Pu et al. (2017). We note that TSI provides greater accuracy and smaller overall uncertainties in the quantification of seasonal snow BC than do thermo-optical carbon analysis (Wang et al., 2020), and thus it has been applied widely in this type of research (Shi et al., 2020). For more detailed information, we refer the reader to Wang et al. (2013b).

## 2.3 Fluorescence and absorption measurement

We obtained absorbance and fluorescence EEMs for filtered meltwater samples via synchronous absorption-3D Fluorescence scanning spectrometry (Aqualog, Horiba Scientific) with the following measurement parameters: excitation = 240–800 nm in 3 nm intervals; emission = 152.25–929.92 nm in 5.04 nm (8 pixels) intervals; scanning interval = 0.3 seconds. Prior to sample measurement, we analyzed aliquots of filtered ultra-pure water (18.2 M ω cm, Milli-q Purification System, Millipore) as analytical blanks. We normalized fluorescence intensity to that of the water Raman unit (RU), which exhibits a peak excitation wavelength of 350 nm, and deducted this Raman signal from all subsequent sample tests (Lawaetz and Stedmon, 2009). The inner filtration effect and Rayleigh scattering peaks were also dispelled following the methods reported by Kothawala et al. (2013) and Bahram et al. (2006), respectively. As fluorescence spectra with wavelengths greater than 600 nm are primarily noise (Zhou et

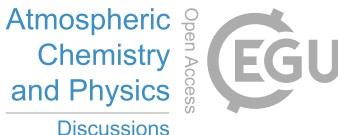

al., 2019b), they are not considered further in this study. Likewise, any samples with absorption spectra
of >600 nm wavelengths were subtracted for the baseline correction (Chen et al., 2019b).
We used version 0.6.3 of the MATLAB drEEM toolbox (http://dreem.openfluor.org/; Murphy et al.,
2013) to perform PARAFAC analysis on EEMs. Comprising the consistency index, residuals, and visual
inspections, the 3-component model is considered more reliable and representative than are the 2–7-
component models (Fig. S1 in the Supplement) and passes the S4C6T3 split scheme (Fig. S2; Murphy
et al., 2013). The contributions of these three components to the overall fluorescence signal are expressed
as relative percentages of $F_{max}$ in RU, and the total fluorescence volume (TFV; RU nm$^2$) is calculated
from the EEMs (Song et al., 2019). Normalized TFV equates to NFV (RU nm$^2$ (mg L$^{-1}$)$^{-1}$), TFV
c(WSOC)$^{-1}$), where c(WSOC) is the concentration of WSOC in the snow (mg L$^{-1}$)), and represents a
sample's fluorescence ability (Chen et al., 2019a).
We calculated three fluorescence-derived indices—the fluorescence index (FI), biological index (BIX),
and humification index (HIX)—from the ratio of fluorescence intensity at specific excitation and
emission wavelengths. As demonstrated by previous studies (Birdwell and Valsaraj, 2010; Huguet et al.,
2009; McKnight et al., 2001), these ratios can help characterize potential sources of WSOC. Specifically,
the FI is taken to represent the relative amount of DOM derived from terrestrial and microbial/algae
sources (McKnight et al., 2001); high values correspond to terrestrially derived organics, and low values
reflect microbial sources. The HIX describes the degree of humification of soluble organic matter
(Zsolnay et al., 1999). During humification, the aromaticity of organic matter increases as microbial
availability decreases, such that higher HIX values correspond to more strongly humified and/or higher
aromaticity organics (principally of terrestrial origin), whereas lower values indicate autochthonous or
microbial origins. As a measure of autochthonous productivity (Huguet et al., 2009), elevated BIX values
are associated with increased contributions of microbial-derived fluorescent organic matter. The three
indices are calculated by the following equations (Ohno, 2002; Huguet et al., 2009; McKnight et al.,
2001; Feng et al., 2016):
$$FI = \frac{I(Ex=370, Em=470)}{I(Ex=370, Em=520)},$$  (1)
$$BIX = \frac{I(Ex=310, Em=380)}{I(Ex=310, Em=430)},$$  (2)
$$HIX = \frac{I(Ex=254, Em=435-480)}{I(Ex=254, Em=300-345)+}$$  (3)



where I is the fluorescence intensity, and Ex and Em represent the excitation and emission wavelengths,
respectively. To ensure a direct comparison with prior results, we recalculated published HIX data using
the same calculation methods as in our own analyses.
We converted sample absorbance to an absorption coefficient using the following equation:
$$a_{WSOC}(\lambda) = ln(10) \cdot Abs(\lambda)/L \qquad (4)$$
where Abs is absorbance, $\lambda$ is wavelength, L is the path length of the cuvette (0.01 m), and $a_{WSOC}$ is the
absorption coefficient ($m^{-1}$).
Owing to the absorption characteristics of WSOC, we selected the absorption coefficient at 280 nm
($a_{WSOC}(280)$) to characterize the absorption intensity of WSOC.
To investigate the wavelength dependence of WSOC absorption, we obtained the Absorption Ångström
exponent (AAE) via the following equation (Doherty et al., 2010; Niu et al., 2018; Wang et al., 2013b;
Yan et al., 2016):
$$a_{WSOC}(\lambda) = K \cdot \lambda^{-AAE} \qquad (5)$$
where K is a constant related to WSOC concentration.
We calculated the mass absorption coefficient ($MAC_\lambda$, $m^2\ g^{-1}$) of our samples using the equation (Chen
et al., 2019b; Yan et al., 2016):
$$MAC_\lambda = a_{WSOC}(\lambda)\ /\ c(WSOC) \qquad (6)$$
where $a_{WSOC}$ is the absorption coefficient derived from Equation (4) and c(WSOC) (mg $L^{-1}$) is the
concentration of WSOC.
**2.4    Snow albedo modeling and radiative forcing calculations**
To establish the radiative effect impact of snowpack WSOC in northeastern China, we used SAMDS to
simulate spectral snow albedo. This model is based on asymptotic radiative transfer theory, which has
been verified by previous studies (Li et al., 2021b; Wang et al., 2017) and described in detail by Wang
et al. (2017), and it involves parameters including solar zenith angle, impurity concentrations, snow
radius, and equivalent particle size. Measured values include the concentration of BC and absorption
coefficients of WSOC. To quantify the influence of pollutants on snow albedo, we assumed a semi-
infinite snow layer and uniform snow grain radii of 100 μm for fresh snow and 1000 μm for old snow,





consistent with previous studies (Pu et al., 2021). With the solar zenith angle fixed at 60°, in line with
our sampling dates and locations, we calculated the reduction in spectral snow albedo for the UV–vis
(280–400 nm) and ultraviolet–near infrared (UV–NIR; 280–1500 nm) bands. Radiative forcing was then
derived by multiplying the albedo reduction value by the incident solar radiation (Painter et al., 2013),
permitting us to evaluate radiative forcing under four scenarios: pure snow, WSOC only, BC only, and
WSOC + BC.
**3    Results and discussion**
**3.1    Characteristics of chemical species**
Figure 2a shows the spatial distribution of measured WSOC in seasonal snow across northeastern China.
Averaged across our entire study area, the mean WSOC concentration (arithmetic mean ± standard
deviation) is $3.59 \pm 3.19$ µg g$^{-1}$, with a maximum of 17.99 µg g$^{-1}$ and a minimum of 0.29 µg g$^{-1}$. Among
the five regions, WSOC concentrations are highest in SNC (average $5.7 \pm 3.68$ µg g$^{-1}$), likely reflecting
the greater degree of agricultural and industrial activity there compared with other regions (Lu et al.,
2011; Wang et al., 2013b). We highlight that both agricultural and industrial sources are considered
anthropogenic. In contrast, our second highest measured concentrations ($3.35 \pm 1.49$ µg g$^{-1}$) are from
SEIM, where desertification occurs (Fang et al., 2007) and is therefore considered a natural source of
WSOC. For most sites, the underlying surface is desert (Fig. 1a) that was incompletely covered by
seasonal snow during the sampling period (Fig. 1d). Consequently, the exposure of natural sandy soils is
a potentially significant contributor of WSOC through aeolian erosion and dry deposition.

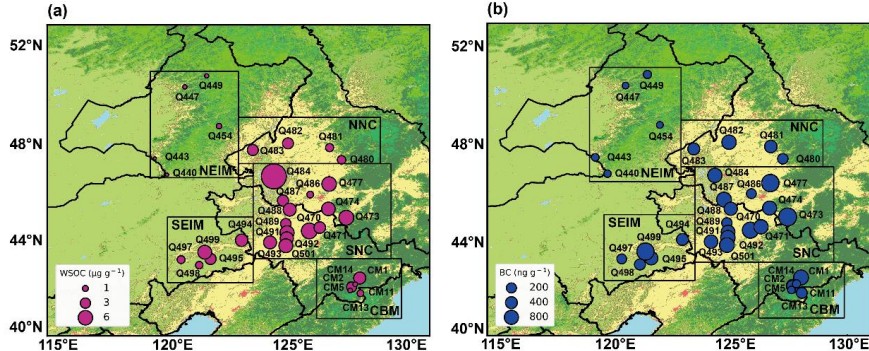

**Figure 2: Spatial distributions of concentrations of (a) WSOC and (b) BC in snow samples. Sampling sites are**
**divided into the five groups defined in Figure 1. Bubble sizes are proportional to concentrations of WSOC**
**and BC.**



In NNC, where both the population density and agricultural intensity are lower than in SNC (Choi et al.,
2020), the contribution of anthropogenic pollution to snowpack is correspondingly lower, resulting in a
moderate WSOC concentration of $2.7 \pm 0.75$ µg g$^{-1}$. Meanwhile, far from intensive human activity, both
CBM and NEIM (Fig. 1a, b, and g) returned low WSOC concentrations (CBM: $1.95 \pm 1.28$ µg g$^{-1}$; NEIM:
$0.50 \pm 0.19$ µg g$^{-1}$). Nonetheless, the higher value for CBM betrays the influence of far-traveled
anthropogenic pollutants (Wu et al., 2020; Zhang et al., 2013).
In comparison with previous studies, we observed that, with the exception of NEIM, our measured
WSOC concentrations are significantly higher than those reported for snow/ice from the Tibetan Plateau
(TGL; ~0.71–1.02 µg g$^{-1}$; Feng et al., 2016), the Alps (~0.14–0.78 µg g$^{-1}$; Vione et al., 2021), North
America (~0.1–0.3 µg g$^{-1}$; Fellman et al., 2015), and polar regions (~0.12–0.27 µg g$^{-1}$; Antony et al.,
2014), but comparable to values in Laohugou glacier ice from the Tibetan Plateau (~1.02–2.6 µg g$^{-1}$;
Feng et al., 2018, 2016) and seasonal snowpack in northwestern China (0.48–7.07 µg g$^{-1}$; Zhou et al.,
2021). This finding implies that snowpack WSOC in northeastern China is contributing significantly to
regional and global climate change (Domine, 2002).
A similar spatial pattern is exhibited by snowpack BC (Fig. 2b). For example, of all five regions, the
regional mean BC concentration is highest for SNC (mean: $922.99 \pm 512.10$ ng g$^{-1}$), followed by SEIM
($659.17 \pm 581.85$ ng g$^{-1}$), NNC ($494.13 \pm 223.81$ ng g$^{-1}$), and the CBM ($391.38 \pm 312.49$ ng g$^{-1}$). BC
concentrations are lowest in NEIM ($59.79 \pm 18.68$ ng g$^{-1}$), in agreement with the values in remote areas
reported by Doherty et al. (2010).
**3.2   Fluorescence characteristics of WSOC**
Three fluorescent components (C1–C3) were captured by resolving the EEMs spectra; all fluorescence
information is summarized in Table S2. C1 exhibits a primary peak at Ex = 240 nm, Em = 448 nm,
indicating a high-oxygenated HULIS found primarily in aromatic conjugated macromolecules (Chen et
al., 2016). The weaker secondary peak occurs at longer excitation wavelengths (Ex / Em = 308 / 448 nm),
implying a higher aromatic content and greater molecular weight (Coble et al., 1998). Wen et al. 2021
concluded that C1 is probably derived from natural terrestrial sources, such as dust and soil, as proposed
originally by Stedmon et al. (2003) and Osburn et al. (2016). Accordingly, we classified C1 as a terrestrial,
humic-like substance, hereafter referred to as HULIS-1.



C2 exhibits a primary (secondary) peak at Ex = 240 (293) nm, Em = 398 nm, suggestive of
lower-oxygenated HULIS (Chen et al., 2016). Observed in a variety of sources, Stedmon et al. (2003)
reported this component in terrestrial end-member samples, whereas both Murphy et al. (2011) and
Osburn et al. (2016) have since linked C2 to anthropogenic sources, such as urban runoff and sewage.
Microbial activity and the degradation of phytoplankton in natural aquatic systems are also thought to
contribute to this component (Yamashita et al., 2008; Zhang et al., 2009). Accordingly, we classified C2
as humic-like substances with complex origins in terrestrial, anthropogenic, and/or microbial sources,
hereafter termed HULIS-2. Unlike HULIS-1 and HULIS-2, C3 is recognizable as a UVB-like protein or
tyrosine-like fluorescence (hereafter PRLIS) with a primary (secondary) peak at Ex = 240 (293) nm, Em
= 398 nm (Osburn et al., 2016; Stedmon and Markager, 2005). PRLIS reflects autochthonously labile
DOM produced by biological processes (Stedmon et al., 2003) and has been reported in previous studies
of seasonal snow (Zhou et al., 2019b).

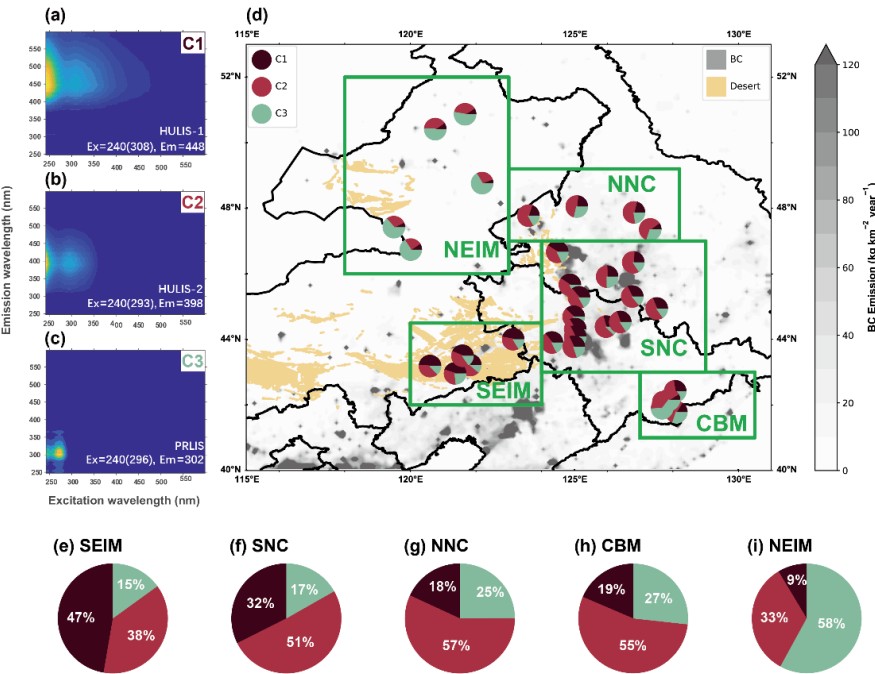

**Figure 3: (a–c) Three fluorescent components identified by PARAFAC analysis. (d) Relative contributions of**
**the three components to total fluorescence at each site. HULIS-1, HULIS-2, and PRLIS are represented by**
**the specific colors shown in the legend (top left corner). The distributions of BC emissions and desert areas in**
**our study area are indicated by gray and light yellow, respectively, with darker gray colors indicating higher**
**black carbon concentrations. (e–i) Average contributions of the three components in different groups of**



**samples. BC emission data are derived from the research group at Peking University**
**(http://inventory.pku.edu.cn/ home.html, Wang et al., 2014a); the Chinese desert (sand) distribution dataset**
**is provided by the National Tibetan Plateau Data Center (http://poles.tpdc.ac.cn/zh-hans/data/122c9ac2-53ee-**
**4b9a-ae87-1a980b131c9b/; Wang et al., 2013a).**
Figure 3d depicts the spatial distribution of the relative contribution of three components to fluorescence,
with the regional averages given in Figure 3e–g. In SEIM, the greatest contribution is that of HULIS-1
(47 %), followed by HULIS-2 (38 %) and PRLIS (15 %), indicating that the signal is dominated by local
soil/dust sources, consistent with the local environment (Figs. 2 and 3d). HULIS-2 plays a greater role
in SNC, where it accounts for half of the total fluorescence signals; of the remaining half, HULIS-1 is
most important. This difference in key components between SEIM and SNC illustrates the change in
primary source of fluorescence intensity. Indeed, with the most intensive human activity (e.g., agriculture,
industrial emissions) being located in SNC (Figs. 1a and 3d; Guo and Hu, 2022), HULIS-2 might be
derived from any combination of terrestrial, anthropogenic, and microbial sources. Nonetheless, in
agreement with previous studies (Zhou et al., 2019b), our combined analysis suggests that anthropogenic
activity is the main contributor to seasonal snow in northeastern China.
As in SNC, HULIS-2 also represents approximately half of the fluorescence signal in both NNC and the
CBM. In the latter, which is heavily forested (Fig. 1a; Guo and Hu, 2022), the dominance of HULIS-2
reflects the long-range transport of anthropogenic pollutants, as discussed in Sect. 3.1. HULIS-1 accounts
for less than PRLIS in both NNC and the CBM, which we posit reflects the concealment of bare soil
surfaces by deep snow and the importance of biological processes due to the heavy vegetation cover.
PRLIS accounts for >50 % of the total fluorescence signal in NEIM, followed by HULIS-2; HULIS-1
contributes relatively little in this region. We attribute this pattern to both the extensive grassland and
forest cover, which obscures bare soil surfaces, and the distance from anthropogenic pollution, which
together serve to amplify the importance of biological processes (Zhou et al., 2019a). Taken as a whole,
the respective contributions of HULIS-1, HULIS-2, and PRLIS to the fluorescence signals in our study
area are ~30 %, ~50 %, and ~20 %. We note that these findings correspond well with the background
environmental conditions.

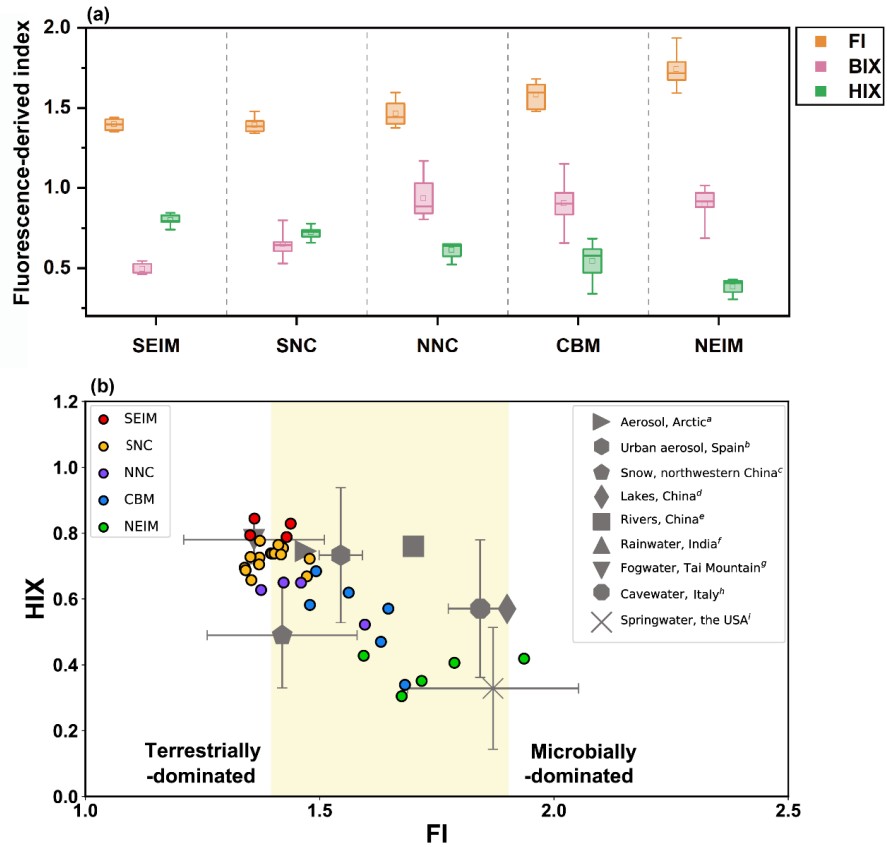

**Figure 4: (a) Variations in fluorescence-derived indices among the five groups. Boxes denote the 25th and 75th**
**quantiles, and horizontal lines represent median values. Averages are shown as small boxes, the whiskers**
**denoting maximum and minimum data. (b) Comparison plots of HIX versus FI for the seasonal snow surface**
**samples (colored dots) from northeastern China, together with the average and standard deviation of different**
**types of WSOC (grey markers) adapted from: Arctic aerosols ([a] Fu et al., 2015), Spanish urban aerosols ([b]**
**Mladenov et al., 2011), seasonal snowpack in northwestern China ([c] Zhou et al., 2019b), Chinese lakes and**
**rivers ([d, e] Zhou et al., 2017), rainwater from Rameswaram, India ([f] Salve et al., 2012), fog water from Tai**
**Mountain, China ([g] Birdwell and Valsaraj, 2010), cave water from Frasassi Caves, Italy ([h] Birdwell and Engel,**
**2010), and spring water in the USA ([i] Birdwell and Engel, 2010). Shaded areas represent mixed WSOC**
**signatures.**
The FI, BIX, and HIX indices reveal spatial variability in fluorescence characteristics and thus permit
the tracing of potential sources. Regionally averaged FI, BIX, and HIX values are depicted in Figure 4a.
Our results show that, in general, FI varies in the range of 1.34–1.94 (mean = 1.49), BIX between 0.46
and 1.17 (mean = 0.74), and HIX between 0.30 and 0.84 (mean = 0.64). By comparison, reported mean
FI, BIX, and HIX values for seasonal snow in Xinjiang (northwestern China) are 1.42, 0.76, and 0.55,





respectively (Zhou et al., 2019b), suggesting that the impact of humification and WSOC aromaticity are
slightly higher in our study area than in Xinjiang. This outcome implies a relatively strong terrigenous
signal and correspondingly weaker biogenic signal in the seasonal snowpack of northeastern China.
Regionally, SEIM exhibits the lowest FI (mean = 1.40) and BIX (mean = 0.49) values but the largest
HIX value (mean = 0.80), further confirming the strong influence of highly aromatic, terrestrially derived
WSOC in this region relative to the others. In contrast, NEIM returns the highest FI (mean = 1.74) and
BIX (mean = 0.89) values, but the lowest HIX value (mean = 0.38), indicating the dominance of low-
aromatic WSOC of microbial origin. Intriguingly, our results reveal that FI and BIX rises monotonously
with decreasing (increasing) fractional contributions of HULIS-1 (PRLIS), whereas HIX exhibits a
similar but contrasting pattern. Together, the comprehensive dataset described above verifies the regional
variability in the terrestrial contributions to WSOC, in which SEIM > SNC > NNC > CBM > NEIM; this
pattern is reversed for microbially sourced WSOC.
Figure 4b illustrates HIX versus FI as a scatterplot, compared with published data for different sample
types; the shaded area depicts the region in which the FI value is >1.4 but <1.9. As FI values of ≤1.4
correspond to terrestrial sources and values of ≥1.9 denote a primarily microbial origin, values of 1.4–
1.9 suggest a mixed origin. In general, FI exhibits a rising trend with declining HIX across northeastern
China. For both SEIM and SNC, FI occupies a narrow range centered on 1.4, indicating either a
predominantly terrestrial or mixed origin. We note that these results are comparable to those of fog water
from the Tai Mountain, Arctic atmospheric aerosols, and seasonal snow in northwestern China (Birdwell
and Valsaraj, 2010; Fu et al., 2015; Zhou et al., 2019b). Further, we highlight that HIX values are
marginally higher in SEIM than elsewhere, suggesting a stronger influence from highly humified WSOC
that probably reflects the extensive deserts and exposed earth in this region. FI values for NNC and the
CBM fall within the range of 1.4–1.7 and thus reflect a mixed origin, in line with previous data from
urban aerosols in Spain and Chinese river water samples (Mladenov et al., 2011, Zhou et al., 2017).
When combined, FI and HIX values for NNC and CBM snowpack indicate that WSOC in these regions
bears a stronger terrestrial signature than do water samples from Chinese lakes and Italian caves
(Birdwell and Engel, 2010; Zhou et al., 2017). Finally, FI values for NEIM fall within a range of 1.6–
2.0, comparable to values from spring water in the USA (Birdwell and Engel, 2010), thus implying a
predominantly microbial or mixed origin.





**3.3   Comparisons of fluorescence and absorption characteristics**
Figure 5a depicts TFV as a measure of the spatial distribution of absolute WSOC fluorescence intensity
in the snowpack of northeastern China; $a_{WSOC}(280)$ is shown in Figure 5b for comparison. In general,
TFV and $a_{WSOC}(280)$ both exhibit large spatial variability in the range of 690–18600 RU nm$^2$ and 0.42–
16.98 m$^{-1}$, respectively. Regional mean values are 7700 ± 2800 RU·nm$^2$ (TFV) and 6.90 ± 2.39 m$^{-1}$
($a_{WSOC}(280)$) for SEIM, 12400 ± 4300 RU·nm$^2$ (TFV) and 11.48 ± 3.96 m$^{-1}$ ($a_{WSOC}(280)$) for SNC, 7800
± 500 RU·nm$^2$ (TFV) and 7.02 ± 0.88 m$^{-1}$ ($a_{WSOC}(280)$) for NNC, 3900 ± 2500 RU·nm$^2$ (TFV) and 3.97
± 2.46 m$^{-1}$ ($a_{WSOC}(280)$) for the CBM, and 1000 ± 300 RU·nm$^2$ (TFV) and 0.83 ± 0.23 m$^{-1}$ ($a_{WSOC}(280)$)
for NEIM. We note that both distributions are consistent in space (Fig. 5e), with the highest
concentrations in SNC and the lowest in NEIM. Moreover, the $a_{WSOC}(280)$ value for SNC is an order of
magnitude larger than that for NEIM, implying that the impact of WSOC on snow albedo at UV
wavelengths is significant in SNC but less notable in NEIM in general (see Sect. 3.5). Previous work has
reported a similarly broad range of snowpack $a_{WSOC}(280)$ (0.15–10.57 m$^{-1}$) in northwestern China (Zhou
et al., 2019b).

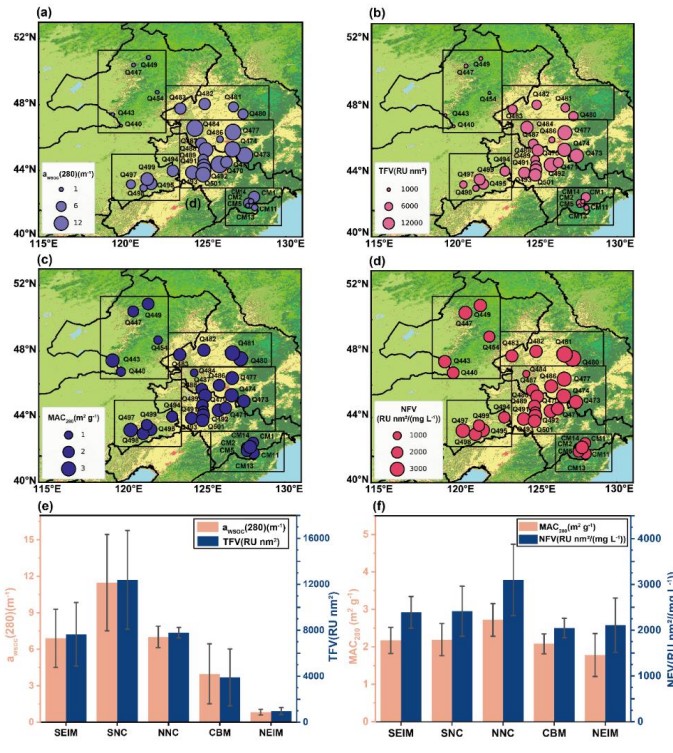





**Figure 5: Spatial distribution of (a) a$_{WSOC}$(280) (m$^{-1}$), (b) MAC$_{280}$, (m$^2$ g$^{-1}$), (c) TFV (RU nm$^2$), and (d) NFV (RU nm$^2$ (mg L$^{-1}$)$^{-1}$). Regional averages for (e) a$_{WSOC}$(280), TFV, (f) MAC$_{280}$ and NFV for the five groups. Error bars in (e) and (f) represent the standard deviations of a$_{WSOC}$(280), MAC$_{280}$, TFV, and NFV for the five groups, respectively.**

Two additional fluorescence and absorption capacity indices, identified as NFV and MAC$_{280}$, are proven tools for revealing WSOC's fluorescence and absorption characteristics and they are related to chemical composition, structure, and source (Chen et al., 2019a). For our study area as a whole, mean NFV and MAC$_{280}$ values are 2411.57 $\pm$ 373.63 RU nm$^2$ (mg L$^{-1}$)$^{-1}$ and 2.17 $\pm$ 0.49 m$^2$ g$^{-1}$, respectively. Both indices exhibit a narrow range, with regional means ranging from 2100 $\pm$ 600 to 3100 $\pm$ 800 RU nm$^2$ (mg L$^{-1}$)$^{-1}$ and from 1.78 $\pm$ 0.57 to 2.72 $\pm$ 0.44 m$^2$ g$^{-1}$, respectively, in contrast to the broad inter-regional disparities in TFV and a$_{WSOC}$(280). Moreover, the spatial patterns of NFV and MAC$_{280}$ are similar, with the highest values in NNC. We speculate that this result reflects the comparatively high low-oxygenated HULIS-2 fraction measured in the NNC samples (Fig. 3g), as the lower-oxygenated BrC (e.g., HULIS-2) has a higher absorption capacity (Browne et al., 2019).

Scatterplots for a$_{WSOC}$(280) with TFV, F$_{max}$(HULIS-1), F$_{max}$(HULIS-2), and F$_{max}$(PRLIS) are provided in Figure S4 to further demonstrate the close relationship between fluorescence and the absorption characteristics of WSOC in our snow samples. As samples Q480, Q484, and Q477 deviate considerably from the respective confidence intervals, we did not include them in our analyses. Surprisingly, we found that TFV is closely correlated to a$_{WSOC}$(280), with P < 0.001 and all datapoints located close to the line of best fit, indicating that the three components (HULIS-1, HULIS-2, PRLIS) contributing to the total fluorescence are also responsible for the majority of absorption. For each component, our data show that F$_{max}$(HULIS-1) is most closely correlated with a$_{WSOC}$(280), followed by F$_{max}$(HULIS-2). The correlation between F$_{max}$(PRLIS) and a$_{WSOC}$(280) is the poorest, yet it is still significant (P < 0.001). Together, our results imply that HULIS-1 is probably the greatest contributor to light absorption, with PRLIS being the least important.

**3.4 Fractional contributions of different WSOC components to light absorption**

Previous studies of atmospheric aerosols, water, and snow/glacier ice have typically regarded WSOC as a whole when discussing its impact on light absorption (Barrett and Sheesley, 2017; D'Sa et al., 2014; Niu et al., 2018; Wu et al., 2019). Yet, depending on environmental conditions, the various components of WSOC play measurably different roles in light absorption according to their concentrations and optical





properties (Zhou et al., 2022). Although the qualitative analysis described above have provided plausible
information about the component-specific influence of WSOC on light absorption, the quantitative
fractional contributions of specific components to light absorption are still unknown. Recently, Chen et
al. (2019a) collected atmospheric aerosol samples in $PM_{2.5}$ over Xi'an, China, and successfully attributed
the dithiothreitol (DTT) activity levels to various BrC components by coupling DDT and BrC datasets.
A similar attribution method has been applied to various research areas, including climate change,
extreme weather, and atmospheric environments (Cao et al., 2015; Pokrovsky, 2019; Xin et al., 2016;
Zhao et al., 2019). In this study, we applied a multiple linear regression method comparable to that of
Chen et al. (2019a) to derive the fractional contribution of each WSOC component to light absorption.
We note that, despite this method having been used elsewhere (Wu et al., 2022; Wu et al., 2021), it
remains a highly innovative approach to evaluating the light absorption of snowpack WSOC.
Table S3 lists the statistical results of the fitted light absorption coefficient, based on the $F_{max}$ data for
three fluorescent components of EEM analysis. As the fitted results can explain ~94 %–99 % of the
variance in measured light absorption within the 280–400 nm range, we conclude that the fusion of
multiple fluorescent components is an effective means of describing most of the spatial features of
WSOC light absorption throughout northeastern China. Accordingly, the wavelength-dependent
fractional contributions of each component of light absorption in this band (280–400 nm) are reported in
Figure 6. For our entire study area, light absorption is dominated by high-oxygenated HULIS-1, which
accounts for ~56 %–65 % of the contribution across UV wavelengths. Further, we observed that the
HULIS-1 contribution rises slightly from 280 to ~340 nm, after which there is a decreasing trend as
wavelength increases. In contrast, HULIS-2 exhibits a valley-type pattern in fractional contribution
between 280 and 400 nm and is responsible for ~19 %–30 % of all light absorption.



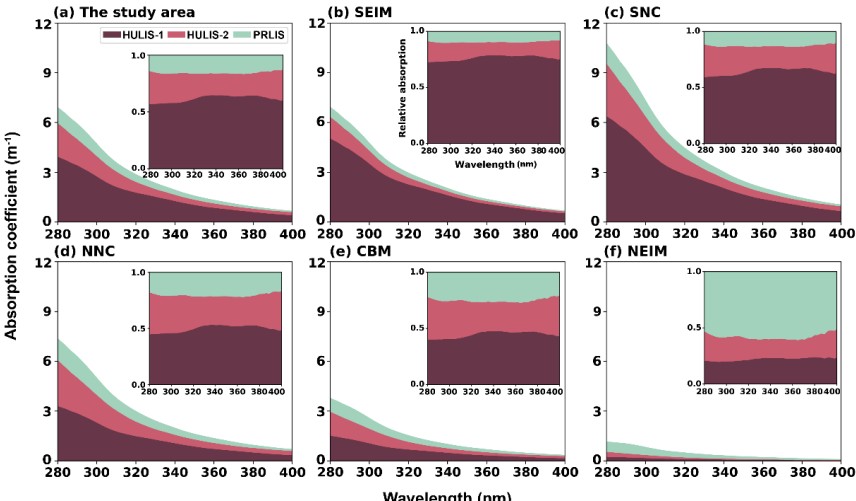

**Figure 6: Relative contributions of the three components to the total absorption of samples in (a) the whole study area and (b–f) each of the five groups.**

PRLIS contributes the least (~12 %–17 %) to light absorption and exhibits a similar wavelength-dependent pattern to HULIS-1. These results are consistent with the qualitatively comparative analysis described in Sect. 3.3. Previous studies have also highlighted this dominance of high-oxygenated compounds in WSOC light absorption, based on samples impacted by naturally and anthropogenically derived soils (Zhou et al., 2022). Conversely, the total absorption coefficient of WSOC decreases with increasing wavelength between 280 and 400 nm, in accord with previous studies (Andreae and Gelencser, 2006; Chakrabarty et al., 2010; Gustafsson et al., 2009; Wu et al., 2019). The AAE lies primarily between 5.0 and 8.0 (mean = 6.6) in the range of 280–400 nm, which is in agreement with results from snow collected from the Arctic, the northern Tibetan Plateau, and northwestern China (Voisin et al., 2012; Yan et al., 2016; Zhou et al., 2021).

For each component, the wavelength-dependent variability in light absorption is similar among all five regions, although the magnitude of each contribution varies from region to region. Moreover, compared with the spectral results, we found that the solar-radiation-weighted broadband light absorption was a more meaningful parameter for researchers studying climate change and atmospheric radiation. Therefore, the broadband results in Figure 7a for 280–400 nm absorption contributions—HULIS-1 (62 %), HULIS-2 (21 %), and PRLIS (17 %)—are average values for the whole study area. On a regional scale, the HULIS-1 contribution to light absorption (280–400 nm) follows the spatial pattern SEIM >

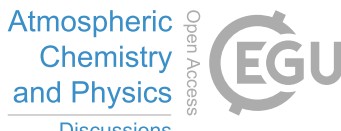

SNC > NNC > CBM > NEIM. We note that HULIS-1 dominates light absorption in SEIM, SNC, and
NNC but has a minor impact in NEIM compared with the other two components. In contrast, the impact
of HULIS-2 varies only slightly among the five regions, with the greatest contributions in NNC and
CMB, and the lowest in NEIM. The contribution of PRLIS is essentially opposite that of HULIS-1, being
dominant in NEIM but of relatively minor important elsewhere. As shown in Figure 7b, light absorption
contributions at 280 nm are consistent with the broadband results (Fig. 7a) in terms of the regional pattern,
although specific values differ because of the different wavelength-dependent properties of light
absorption for the three WSOC components.

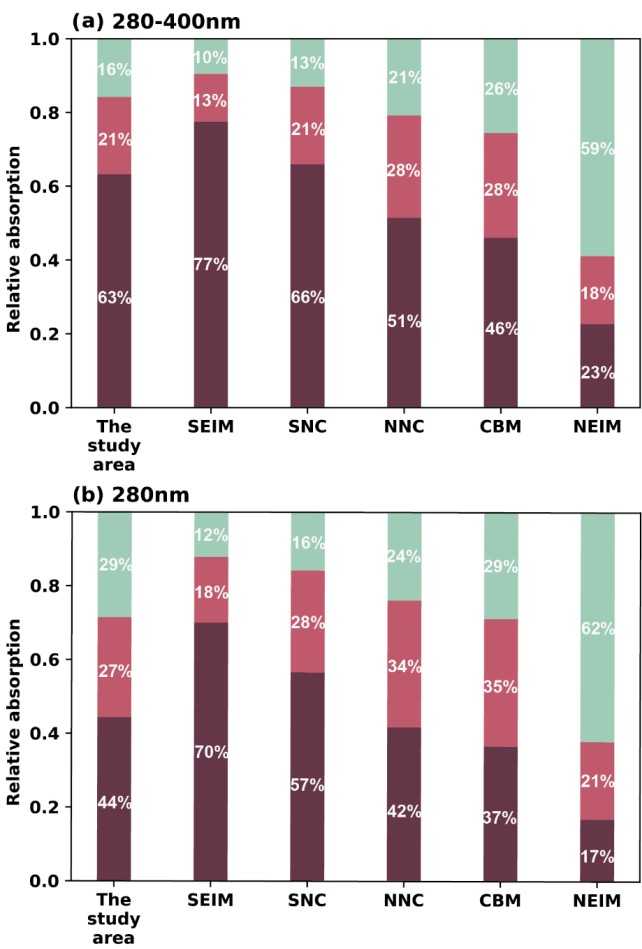

**Figure 7: Regional averages for the relative contributions of the three fluorescent components to light**
**absorption at wavelengths of (a) 280 nm and (b) 280–400 nm.**



We find it noteworthy that, for each component, the overall regional pattern of its contribution to light
absorption aligns with its impact on fluorescence signals, thereby confirming the viability of the
attribution analysis employed in our study. Nonetheless, we observed that the magnitude of each
component's contribution varies relative to its respective fluorescence signal. For instance, HULIS-1
returns a greater contribution to light absorption than its fluorescence signal, in contrast to HULIS-2.
One plausible explanation for this discrepancy is that the fluorescence quantum yields (AQYs), which
are essentially the ratio of fluorescence intensity versus absorption intensity, are different for each
component. Indeed, in their comprehensive field-based study of BrC fluorescence and absorption
properties in northern China, Wen et al. (2021) reported that the AQYs of WSOC decrease with
increasing HIX, meaning that components with higher HIX values, such as HULIS-1, have lower AQYs
than does HULIS-2. Thus, the contribution of HULIS-1 to the fluorescence signals will be smaller than
its contribution to light absorption, and vice versa for HULIS-2.
**3.5   Albedo reduction and radiative forcing attributed to snowpack WSOC**
The strong light absorption of WSOC in UV bands has important ramifications for snow albedo and
radiative forcing throughout northeastern China. However, owing to the chemical and optical complexity
of WSOC components, quantitative estimates for snowpack light absorption remain poorly understood.
For example, although prior work in northeastern China has focused on BC (Wang et al., 2013b) and
other water-insoluble light-absorbing particles (Wang et al., 2017; Zhao et al., 2014) via field
measurements, model simulations, and satellite remote sensing (Pu et al., 2019), the specific impacts of
WSOC have not been studied. Consequently, ours is the first study to report on the impact of WSOC on
snow albedo and radiative forcing in northeastern China and to compare these data with BC results to
highlight the non-negligible role of WSOC.



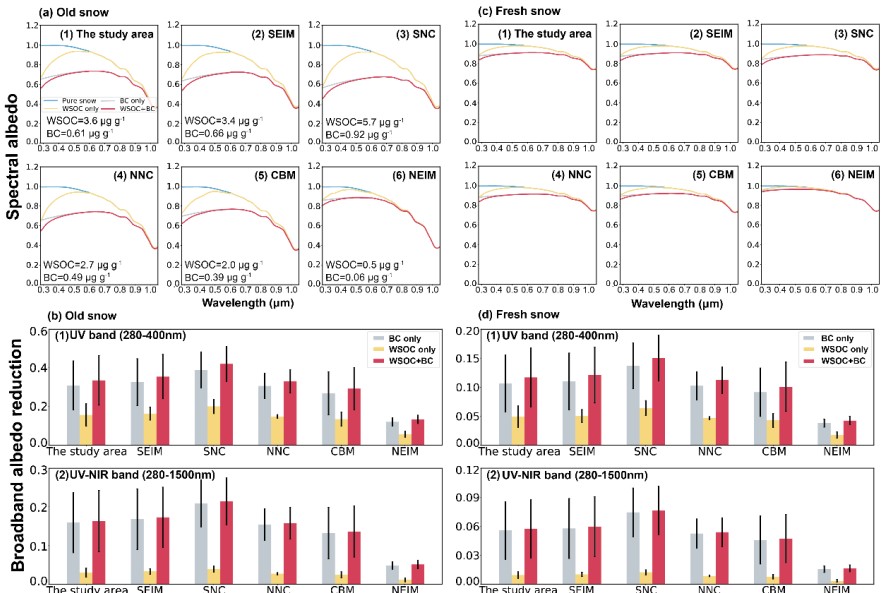

**Figure 8: (a) and (b): Simulated snow spectral albedo and broadband albedo reductions—under various contamination scenarios and for different regions—for old snow (radius = 1000 μm). (c), (d) Simulated snow spectral albedo and broadband albedo reductions—under various contamination scenarios and for different regions—for fresh snow (radius = 100 μm). Colors represent the different types of snow (pure snow, BC- or WSOC-contaminated snow, and snow polluted by both WSOC and BC).**

Figure 8 shows the regional-mean spectral snow albedo as well as the reduction in albedo due to WSOC, BC, and WSOC + BC. We assume a snow radius of 100 μm for fresh snow and 1000 μm for old snow. Our findings reveal that WSOC induces a marked decline in albedo within the UV and short-wave VIS bands, with the magnitude of albedo reduction growing rapidly as wavelength shrinks owing to the large AAE value of WSOC. In comparison, BC induces a widespread albedo reduction spanning the UV to NIR bands, and wavelength-dependent variations are significantly smaller than those of WSOC. For VIS and NIR, the reduction in albedo is dominated by BC, whereas the impacts of WSOC and BC are comparable in UV wavelengths, a pattern that is consistent with the results of studies of atmospheric aerosols (Shamjad et al., 2016). We note that these characteristics persist throughout northeastern China despite regional variability in environmental conditions and snowpack types (old vs. fresh snow).

For broadband wavelengths, our results indicate that the WSOC-induced (mean = 3.6 μg g$^{-1}$) albedo reduction for 280–400 nm wavelengths in old (fresh) snow is 0.16 (0.05) across the whole study area, which corresponds to approximately 50.3 % (46.3 %) the impact of BC (mean = 0.6 μg g$^{-1}$). Regionally,





the greatest decline in albedo occurred in SNC, where a mean WSOC of 5.7 µg g$^{-1}$ resulted in a reduction
of 0.20 (0.06) in the 280–400 nm range for old (fresh) snow. In contrast, the smallest decline in albedo
was observed in NEIM, with reductions of 0.06 (0.02) resulting from an average WSOC concentration
of 0.5 µg g$^{-1}$. Compared with the UV bands, a WSOC-induced albedo reduction of 0.03 (0.009) over the
UV–NIR range (280–1500 nm) accounts for only ~18.8 % (16.7 %) of that due to BC in our study area.
The regional mean for old (fresh) snow falls in the range of 0.01–0.04 (0.003–0.012), with the highest
(lowest) values occurring in SNC (NEIM). However, we observed the highest ratio of WSOC- to BC-
induced albedo reduction in NEIM. Together, these results indicate that WSOC plays a potentially
important role in altering UV snow albedo in NEIM, despite its relatively low concentrations in the
regional snowpack.

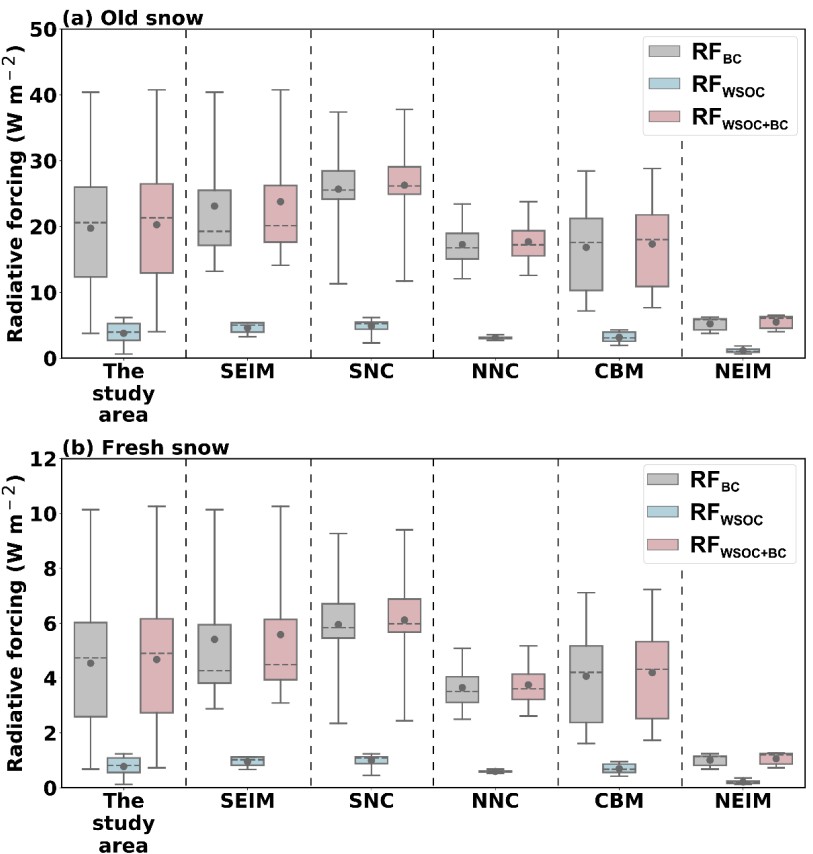

**Figure 9: Radiative forcing due to different pollutants in (a) old or (b) fresh snow. Gray, blue, and red indicate**
**the radiative forcing of BC, WSOC, and BC + WSOC, respectively.**



Radiative forcing is an important index that directly reflects the impact of snowpack WSOC on the
regional radiation balance and climate (Beres et al., 2020). Previous studies have tended to focus on
calculating instantaneous radiative forcing values; however, in reality, time-averaged results are more
valuable for climate research. Here, we present data on the daily mean radiative forcing due to WSOC,
BC, and WSCO + BC (Fig. 9), calculated using the methodology of Wang et al. (2017). In general, for
northeastern China we found the mean radiative forcing of WSOC in old (fresh) snow to be 3.78 (0.77)
W m$^{-2}$, with regional mean values varying from 1.15 (0.21) to 4.88 (1.0) W m$^{-2}$. Zhou et al. (2021)
reported daily mean radiative forcing by regional WSOC (0.6–7.1 µg g$^{-1}$) of between ~0.04 and ~0.59
W m$^{-2}$ for northwestern China, which is comparable to our values in fresh snow. Furthermore, the ratio
of WSOC-driven to BC-driven radiative forcing varies within the range of 10.3 %–32.0 % (9.8 %–30.8 %)
for old (fresh) snow, which is consistent with the results of our calculated albedo reductions. These results
confirm that the role of WSOC must not be ignored in discussions about radiative balance in northeastern
China. Similarly, the sizeable impact of WSOC on the absorption of UV radiation has the potential to
influence biogeochemistry (Helms et al., 2013; Seekell et al., 2015) and snow photochemical processes
(e.g., photolysis of nitrate ($NO_3^-$) and nitrite ($NO_2^-$) in snow, in addition to the release of NOx (NO +
$NO_2$ and HONO). Snow photochemistry is beyond the scope of this study, however, the high
concentrations of WSOC and nitrate (not shown) pollution in northeastern China make this a logical next
step for research in this field.
**4    Conclusions and atmospheric implications**
During 2020 and 2021, we collected 34 surface samples of seasonal snow from sites throughout
northeastern China to investigate the fluorescence characteristics, optical properties, and radiative effects
of snowpack WSOC. With an average concentration of WSOC of 3.59 ± 3.19 µg g$^{-1}$, our results returned
regional mean values of 3.35 ± 1.49 µg g$^{-1}$ (SEIM), 5.73 ± 3.68 µg g$^{-1}$ (SNC), 2.70 ± 0.75 µg g$^{-1}$ (NNC),
1.95 ± 1.28 µg g$^{-1}$ (CBM), and 0.50 ± 0.19 µg g$^{-1}$ (NEIM), indicating a considerable degree of regional
variability of WSOC mass loadings. Measured values of WSOC fluorescence intensity (690–18600 RU
nm$^2$) and light absorption (0.4–17.0 m$^{-1}$) are also highly variable.
In the first study of its kind, we used EEMs and PARAFAC to identify three fluorescence WSOC
components prevalent in northeastern China. Specifically, these include the high-oxygenated HULIS-1,



which is a terrigenous, humic-like component, and the low-oxygenated HULIS-2, which is a humic-like
component derived from mixed sources, such as anthropogenic activity, microbial processes, and soil.
The third component, PRLIS, is a protein-like substance derived from autochthonous biological activity.
In SEIM, which is characterized by desert and bare soil surfaces, the HULIS-1 signal is dominant (47 %)
and the HIX value is the highest, whereas FI and BIX are the lowest. Together, these findings reveal a
high degree of humification and minimal bioavailability, indicating that snowpack WSOC originates
primarily from soil sources. In contrast, the PRLIS signal (58 %) dominates in NEIM, which also exhibits
the lowest HIX values and highest FI and BIX values. We propose that the elevated bioavailability of
this remote, forested region indicates a predominantly biological origin for NEIM snowpack WSOC.
HULIS-2 dominates the densely populated and intensively farmed SNC (51 %) and NNC (57 %) regions,
where HIX, FI, and BIX values are moderate, leading us to conclude that snowpack WSOC is of mixed
origin.
We employed multiple regression analysis to estimate the fractional contributions of different WSOC
components to snowpack light absorption. Throughout our study area, HULIS-1 tends to be the greatest
contributor (~56 %–65 %) over the 280–400 nm range, followed by HULIS-2 (~19 %–30 %) and PRLIS
(~12 %–17 %). On a more regional basis, light absorption remains dominated by HULIS-1 in SEIM,
SNC, NNC, and the CBM, whereas PRLIS takes a leading role in NEIM. In contrast to its primary role
in fluorescence, the contribution of HULIS-2 to light absorption is relatively low across all regions,
potentially reflecting the variable fluorescence quantum yields (AQYs) of the different components.
Finally, we compared the impact on snow albedo and radiative forcing of WSOC relative to BC. With
an average concentration of 3.6 $\mu g\ g^{-1}$, our results indicate that WSOC induces an albedo reduction
across northeastern China of 0.16 (0.05) in old (fresh) snow over the 280–400 nm range, and thus
represents approximately 50 % (46 %) of the albedo reduction due to BC (average concentration = 0.6
$\mu g\ g^{-1}$). We note, however, that the WSOC-driven reduction in the UV–NIR spectrum (280–1500 nm) is
only 0.03 (0.009), corresponding to 19 % (17 %) that of BC. The average radiative forcing of WSOC in
old (fresh) snow in northeastern China is 3.8 (0.8) W $m^{-2}$, equating to 19 % (17 %) of the BC-induced
radiative forcing.
We indicate that our study could contribute to the understanding of carbon cycling processes, regional
air quality, hydrological processes, and climate change in the earth systems. For example, the abundant



WSCO concentrations measured in this study implied the significant carbon input from the
atmosphere to the snowpack through wet or dry depositions in northeastern China. While the complex
chemical compositions of snowpack WSOC could further influence the carbon balance of the snow
environment by affecting microbial activities (Stedmon et al., 2007). The strong absorption properties of
WSOC in the UV-Vis band also implied its important role in initiating snow photochemistry (McNeill
et al., 2012), which will change the composition of organic compounds in the snow in turn (Grannas et
al., 2007), and affect the surrounding air quality by releasing oxidizing gas like NOx into the atmosphere
(Zatko et al., 2013). Moreover, the non-negligible influence of WSOC on the snow albedo and radiative
effect indicated that it could not only accelerate snow melting, change the periods and mass of water and
carbon exchange between snowpack and underlying soils or vegetation (Meyer and Wania, 2008) , but
also potentially affect regional climate through changing the surface radiative balance (Beres et al., 2020) .
*Data availability*. Data presented and used throughout this study can be accessed through the
following data repository: https://doi.org/10.5281/zenodo.6541956.
*Supplement*. The supplement related to this article is available online at:
*Author contributions*. XN and WP designed the study and wrote the first draft with contributions
from all coauthors. XN designed and conducted the lab experiments with the assistance of YZ and
HW. XN processed the data with the assistance of DW and TS. XN, WP, YC, YX, TS designed
and conducted the field campaign. XW supervised this study. All co-authors commented on the
paper and improved it.
*Competing interests*. The authors declare that they have no conflict of interest.



*Acknowledgements*. The Lanzhou University group acknowledges support from the National
Science Fund for Distinguished Young Scholars and the National Natural Science Foundation of
China.

*Financial support*. This research was supported by the National Science Fund for Distinguished
Young Scholars (42025102), the National Natural Science Foundation of China (42075061 and
41975157), and the Lanzhou City's scientific research funding subsidy to Lanzhou University.

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
