# Peer review of "radiative effects of water-soluble organic carbon in"

_Atmospheric Chemistry and Physics, 2022_

## Author Comment (AC1)

**Response to Editor Prof. Irena Grgić**

Thank you for giving us an opportunity to revise and improve the quality of the manuscript. We have carefully read the reviewers' comments and have made revisions which marked in red in the paper. At the same time, we have responded to the opinions of the two reviewers point by point. To make the reply more visible, our responses start with "R:". The specific reply is as follows:

**Response to Anonymous Referee #1**

Received and published: 19 July 2022

This study presents detailed investigations on the fluorescence characteristics, absorption properties, and radiative effects of water-soluble organic carbon in seasonal snow across northeastern China, which plays a profound effect on snow albedo and solar radiation balance. They found that the WSOC mainly has mixed origins, including anthropogenic activity, microbial activity, and soil. The albedo reduction by WSOC (average concentration of 3.6  $\mu$ g g-1) was about 50% of the albedo reduction due to BC (average concentration of 0.6  $\mu$ g g-1) in the UV-vis (Ultraviolet-visible) band, and the radiative forcing was 3.8 (0.8) W m-2 in old (fresh) snow, which was equal to 19% (17%) of the radiative forcing by BC. The manuscript is generally well-written and presents comprehensive high-quality data analysis, which will be of interest to the community. I believe the manuscript can be accepted after addressing the following comments. R: We are very grateful for reviewer's constructive comments, which have helped us

R: We are very grateful for reviewer's constructive comments, which have helped us improve the paper quality substantially.

 In page 2, Line 12, more references should be given, especially for organic carbon and biota aerosols on the principal LAPs in snow.

R: Revised as suggested. (See Page 2, line 22)

- Beres, N. D., Sengupta, D., Samburova, V., Khlystov, A. Y., and Moosmüller, H.: Deposition of brown carbon onto snow: changes in snow optical and radiative properties, Atmos. Chem. Phys., 20, 6095–6114, https://doi.org/10.5194/acp-20-6095-2020, 2020.
- Els, N., Greilinger, M., Reisecker, M., Tignat-Perrier, R., Baumann-Stanzer, K., Kasper-Giebl, A., Sattler, B., and Larose, C.: Comparison of Bacterial and Fungal Composition and Their Chemical Interaction in Free Tropospheric Air and Snow Over an Entire Winter Season at Mount Sonnblick, Austria, Front. Microbiol., 11, https://doi.org/10.3389/fmicb.2020.00980, 2020.
- Wu, G., Cong, Z., Kang, S., Kawamura, K., Fu, P., Zhang, Y., Wan, X., Gao, S., and Liu, B.: Brown carbon in the cryosphere: Current knowledge and perspective,

*Adv. Clim. Chang. Res.*, 7, 82–89, https://doi.org/10.1016/j.accre.2016.06.002, 2016.

2. I suggested that the first paragraph should be reconstructed; the author should primarily define LAPs and then illustrate their effects on snow albedo reduction.

R: Reconstructed as suggested. The definition of LAPs and their effects on snow albedo reduction have been added as follows in Page 2, line 21-22 to Page 3, line 1-5.

"The LAPs in seasonal snow, such as black carbon (BC), organic carbon (OC), mineral dust (MD), and biota (Beres et al., 2020; Di Mauro, 2020; Els et al., 2020; Qian et al., 2015; Wu et al., 2016), can strongly absorb solar radiation, which together serves to lower surface albedo and impose a positive radiative forcing (Cui et al., 2021; Dumont et al., 2014; Hansen and Nazarenko, 2004; Shi et al., 2022a; Warren and Wiscombe, 1980; Zhang et al., 2017). Ultimately, LAPs can accelerate snow melting (Li et al., 2021b) and disturb the global radiative balance, therefore, have important implications for regional and global climate change (Skiles et al., 2018; Shi et al., 2022b)."

 Page 3, line 8-12, the sentence should be divided into separate sentences. Moreover, Page 3, line 16, references also should be provided after illustrating new investigations.

R: This sentence has been rewritten as follows. (See Page 4, line 11-14)

"Beine et al. (2011) reported that WSOC occupied almost the entire absorption spectrum of the photochemically active region (300–450 nm) in surficial snow samples from Barrow, Alaska. And Feng et al. (2016) observed that absorption in cryoconite samples from the central Tibetan Plateau was dominated by WSOC components in the 300–350 nm range."

The following references are added as suggested (See Page 4, line 18-19).

Li, X., Fu, P., Tripathee, L., Yan, F., Hu, Z., Yu, F., Chen, Q., Li, J., Chen, Q., Cao, J., and Kang, S.: Molecular compositions, optical properties, and implications of

dissolved brown carbon in snow/ice on the Tibetan Plateau glaciers, Environ. Int., 164, 107276, https://doi.org/10.1016/j.envint.2022.107276, 2022.

- Guo, B., Li, W., Santibáñez, P., Priscu, J. C., Liu, Y., and Liu, K.: Organic matter distribution in the icy environments of Taylor Valley, Antarctica, Sci. Total Environ., 841, 156639, https://doi.org/10.1016/j.scitotenv.2022.156639, 2022b.
- 4. Page 4, line 1-11, Atmospheric particulate deposition is an important source of organic matter in snow. The author should expound the application and progress of EEM method in atmospheric particulate matter research. Such as chromophores type identification, origin analysis, atmospheric chemical processes.

R: Thanks for your suggestion. The following content has been added. (See Page 5, Line 8-14)

"At present, the application of EEMs on atmospheric aerosols has entered a mature stage. EEMs have been used to identify fluorescent WSOC components in aerosols from polar regions or urban backgrounds, and it has been found that different structures of WSOC fractions exhibit different oxidation properties, which may provide a clue to understand the chemical formation or loss of organic chromophores in atmospheric aerosols (Chen et al., 2016; Fu et al., 2015). Recently, this method has been gradually extended to the analysis of glacier samples and snow samples (Feng et al., 2018; Guo et al., 2022; Zhou et al., 2019b)."

5. Page 6, line 11-12, the TSI technique was not found in the literature of Shi (2020) and Wang (2014b), please cite with Wang et al (2020), and also provide in the reference list.

R: Revised as suggested. (See Page 9, Line 19)

6. Page 8, line 22-25, the sentence is too long; please divide it into compact sentences. Also carefully revised all other long sentences throughout the manuscript.
R: This sentence has been rewritten as follows. (See Page 13, Line 13-18) *"This model is based on asymptotic radiative transfer theory, which has been* verified by previous studies (Li et al., 2021b; Wang et al., 2017). As described in detail by Wang et al. (2017), the model involves parameters including solar zenith angle, impurity concentrations, absorption coefficient of impurities, and snow grain radius."

Moreover, other long sentences have also been simplified throughout the manuscript.

 The authors should carefully revise the number of significant digits (e.g. Page 10, line 16-17).

R: The number of significant digits has been carefully revised throughout the manuscript.

- 8. Page 10, line 25, the year of the citation should be given in brackets.R: The citation format has been corrected as suggested. (See Page 16, Line 20)
- 9. Page 11, line 2-4, the sentence should be divided into individual sentences.
  R: This sentence has been rewritten as follows. (See Page 17, line 2-4)
  "Observed in a variety of sources, Stedmon et al. (2003) reported this component in terrestrial end-member samples. Whereas, both Murphy et al. (2011) and Osburn et al. (2016) have since linked C2 to anthropogenic sources, such as urban runoff and sewage."
- 10. Reference to previous studies to explain what HULIS-1 and HULIS-2 are, and clarify the advantages and disadvantages.

R: HULIS-1 is a terrestrial-derived humic fluorophore with a long emission wavelength, commonly reported for samples from terrestrial aquatic systems and highly-oxygenated organic aerosols (Chen et al., 2016; Stedmon et al., 2003). This fluorophore mainly absorbs light in the UVC band and shows a broad emission peak. HULIS-2 with low oxidative properties was first measured in marine systems (Coble, 1996), which is usually considered as HULIS from the ocean (Coble, 1996)

or phytoplankton degradation in freshwater (Zhang et al., 2010). Stedmon et al. (2003) found a similar fluorophore in a terrestrially dominated estuary region. It is also found in anthropogenic wastewater (Stedmon and Markager, 2005) and aerosols of industrial origin (Chen et al., 2020). Thus, in this study, HULIS-1 can be used to represent terrestrial sources, and HULIS-2 can characterize mixed sources including terrestrial sources, anthropogenic sources and microbial activity sources.

The content above corresponds with the Section 3.2.

11. In Figure 3f-g, there is a similar contribution to HULIS-2, but the explanation is still unclear. Please explain.

R: Figure 3f-g show the contribution of fluorescence signals of three components in SNC and NNC, respectively. The background environment of these two regions is similar to the dense urban and population distribution, so they are more vulnerable to anthropogenic activities than other areas. Therefore, the relative contribution of HULIS-2, which represents anthropogenic sources, is the largest, and there is no obvious difference. The regional difference of the fluorescent components in the two regions is mainly reflected in the proportion of the other two components.

The above relevant contents have been supplemented in the manuscript. (See Page 19, Line 10-11)

12. It's better to provide the equations on radiative forcing than that only cited from other references in section 2.4.

R: Revised as suggested. (See Page 13, Line 21-22 to Page 14, Line 1-11) "With the solar zenith angle fixed at 60°, consistent with our sampling dates and locations, we calculated the reduction ( $\Delta a_i$ , i represents BC only, WSOC only, and BC + WSOC, similarly hereinafter.) in spectral snow albedo derived from different types of impurities for the UV–vis (280–400 nm) and ultraviolet–near infrared (UV– NIR; 280–1500 nm) bands. Radiative forcing resulting from either BC or WSOC in snow  $(RF_i)$  was then derived by multiplying the albedo reduction value by the incident solar radiation (Painter et al., 2013):

$$RF_i = E \cdot (\alpha_{pure} - \alpha_i) = E \cdot \Delta \alpha_i \tag{7}$$

where  $a_{pure}$  is snow albedo for pure snow and E is the average daily downward shortwave solar radiation flux acquired from NASA's Clouds and the Earth's Radiant Energy System (CERES) product "CERES SYN1deg" (https://ceres.larc.nasa.gov/products.php?product=SYN1deg)."

13. Compared with the integrated relative absorption from 280 to 400 nm in Fig. 7a, what does Fig. 7b used for? Please explain.

R: The previous study used the absorption coefficient of 280 nm to characterize the absorption (Zhang et al., 2010). Therefore, the results of the absorption contribution for this parameter were given to facilitate comparison with other studies. The explanation above has also been added to the manuscript. (See Page 12, Line 19-21)

- 14. In the conclusion section from Page 23, line 20-Page 24, line 27, parts of the contents were duplicated with the result section. Suggested reconstructed.
  - R: Reconstructed as suggested (See Section 4).

"During 2020 and 2021, we collected 34 surface samples of seasonal snow from sites throughout northeastern China to investigate the fluorescence characteristics, optical properties, and radiative effects of snowpack WSOC. With an average concentration of WSOC of  $3.6 \pm 3.2 \ \mu g \ g^{-1}$ , our results returned regional mean values of  $3.4 \pm 1.5 \ \mu g \ g^{-1}$  (SEIM),  $5.7 \pm 3.7 \ \mu g \ g^{-1}$  (SNC),  $2.7 \pm 0.8 \ \mu g \ g^{-1}$  (NNC),  $2.0 \pm 1.3 \ \mu g \ g^{-1}$  (CBM), and  $0.5 \pm 0.2 \ \mu g \ g^{-1}$  (NEIM), indicating a considerable degree of regional variability of WSOC mass loadings. Measured values of WSOC fluorescence intensity (690–18600 RU nm2) and light absorption (0.4–17.0 m-1) are also highly variable. Moreover, we also used EEMs and PARAFAC to identify three fluorescence WSOC components prevalent in northeastern China, and analyzed their regional differences. In SEIM, which is characterized by desert and bare soil surfaces, the signal of high-oxygenated and terrigenous HULIS-1 is dominant (47 %). The high degree of humification and minimal bioavailability of WSOC, indicating that snowpack WSOC originates primarily from soil sources. In contrast, the autochthonous PRLIS signal (58 %) dominates in remote and clean NEIM. Low-oxygenated and anthropogenic HULIS-2 dominates the densely populated and intensively farmed SNC (51 %) and NNC (57 %) regions, leading us to conclude that the snowpack WSOC in SNC and NNC are influenced more by anthropogenic source. In CBM of forest environment, the impact of long-distance transport of pollutants is greater than that of the background environment. The above conclusions are also verified by fluorescence-derived indices.

We employed multiple regression analysis to estimate the fractional contributions of different WSOC components to snowpack light absorption. Throughout our study area, HULIS-1 tends to be the greatest contributor (~56 %–65 %) over the 280– 400 nm range, followed by HULIS-2 (~19 %–30 %) and PRLIS (~12 %–17 %). In contrast to its primary role in fluorescence, the contribution of HULIS-2 to light absorption is relatively low across all regions, potentially reflecting the variable molecular structure of different components. Finally, we highlighted that the average RF due to WSOC in old (fresh) snow in northeastern China is 3.8 (0.8) W  $m^{-2}$ , which is equal to 19 % (17 %) of the BC-induced radiative forcing. Therefore, we demonstrated the important impacts of WSOC on the snow energy budget and potentially on triggering snow photochemistry."

**References:**

Chen, Q., Miyazaki, Y., Kawamura, K., Matsumoto, K., Coburn, S., Volkamer, R., Iwamoto, Y., Kagami, S., Deng, Y., Ogawa, S., Ramasamy, S., Kato, S., Ida, A., Kajii, Y., and Mochida, M.: Characterization of Chromophoric Water-Soluble Organic Matter in Urban, Forest, and Marine Aerosols by HR-ToF-AMS Analysis and Excitation–Emission Matrix Spectroscopy, Environ. Sci. Technol., 50, 10351–10360, https://doi.org/10.1021/acs.est.6b01643, 2016.

- Chen, Q., Li, J., Hua, X., Jiang, X., Mu, Z., Wang, M., Wang, J., Shan, M., Yang, X., Fan, X., Song, J., Wang, Y., Guan, D., and Du, L.: Identification of species and sources of atmospheric chromophores by fluorescence excitation-emission matrix with parallel factor analysis, Sci. Total Environ., 718, 137322, https://doi.org/10.1016/j.scitotenv.2020.137322, 2020.
- Coble, P. G.: Characterization of marine and terrestrial DOM in seawater using excitation-emission matrix spectroscopy, Mar. Chem., 51, 325–346, https://doi.org/10.1016/0304-4203(95)00062-3, 1996.
- Stedmon, C. A. and Markager, S.: Resolving the variability in dissolved organic matter fluorescence in a temperate estuary and its catchment using PARAFAC analysis, Limnol. Oceanogr., 50, 686–697, https://doi.org/10.4319/lo.2005.50.2.0686, 2005.
- Stedmon, C. A., Markager, S., and Bro, R.: Tracing dissolved organic matter in aquatic environments using a new approach to fluorescence spectroscopy, Mar. Chem., 82, 239–254, https://doi.org/10.1016/S0304-4203(03)00072-0, 2003.
- Zhang, Y., Zhang, E., Yin, Y., van Dijk, M. A., Feng, L., Shi, Z., Liu, M., and Qina,
  B.: Characteristics and sources of chromophoric dissolved organic matter in lakes of the Yungui Plateau, China, differing in trophic state and altitude, Limnol. Oceanogr., 55, 2645–2659, https://doi.org/10.4319/lo.2010.55.6.2645, 2010.

**Anonymous Referee #2**

**27 Jul 2022**

The manuscript discussed the fluorescence characteristics, absorption properties, and radiative effects of WSOC based on 34 snow samples collected from sites in northeastern China. Given that the fluorescence characteristics, absorption properties, and radiative effects of WSOC in snow samples in northeastern China are not addressed in the literature, the work therefore is novel and would add useful information to the climate effect of WSOC. There are some issues and comments that need to be considered before publication:

R: We are grateful for the reviewer's insightful comments, which are helpful and valuable for improving our manuscript. We have addressed all the comments carefully as detailed below in our point-by-point responses.

Specific comments:

 Page 3, Line 2: "the majority of which is water soluble organic carbon (WSOC).", Please provide references.

R: The following references are added as suggested. (See Page 3, Line 22)

- Hood, E., Battin, T. J., Fellman, J., O'Neel, S., and Spencer, R. G. M.: Storage and release of organic carbon from glaciers and ice sheets, Nature Geosci., 8, 91–96, https://doi.org/10.1038/ngeo2331, 2015.
- Yan, F., Kang, S., Li, C., Zhang, Y., Qin, X., Li, Y., Zhang, X., Hu, Z., Chen, P., Li, X., Qu, B., and Sillanpää, M.: Concentration, sources and light absorption characteristics of dissolved organic carbon on a medium-sized valley glacier, northern Tibetan Plateau, The Cryosphere, 10, 2611–2621, https://doi.org/10.5194/tc-10-2611-2016, 2016.
- Page 3, Line 2-6, Some more important information on WSOC should be involved in the Introduction.

R: We have added more information on WSOC. (See Page 3, Line 20-22 to Page 4, Line 1-6)

"The recent study has reported that the storage of OC in mountain glaciers and ice caps (~11 % of Earth's land surface) could be as high as 6 petagrams (Pg), the majority of which is water-soluble organic carbon (WSOC) (Hood et al., 2015; Yan et al., 2016). The substantial part of WSOC in glacier is highly bioavailable and can be a source of labile organic matter for downstream ecosystems (Singer et al., 2012). The physical and photochemical processes can occur within various WSOC in snow cover and glaciers, and therefore have a great effect on atmospheric and glacier chemistry (Domine, 2002; Grannas et al., 2007; Antony et al., 2011). Moreover, WSOC has important influences on the energy budget and radiative forcing of snow cover and glaciers (Kirillova et al., 2014; Ram et al., 2010; Yan et al., 2016)."

3. Page 3, Line 19: Please provide the corresponding references.

R: The following references are added as suggested. (See Page 5, Line 1-2)

- Barrett, T. E. and Sheesley, R. J.: Year-round optical properties and source characterization of Arctic organic carbon aerosols on the North Slope Alaska, J. Geophys. Res.-Atmos., 122, 9319–9331, https://doi.org/10.1002/2016JD026194, 2017.
- D'Sa, E. J., Goes, J. I., Gomes, H., and Mouw, C.: Absorption and fluorescence properties of chromophoric dissolved organic matter of the eastern Bering Sea in the summer with special reference to the influence of a cold pool, Biogeosciences, 11, 3225–3244, https://doi.org/10.5194/bg-11-3225-2014, 2014.
- Niu, H., Kang, S., Lu, X., and Shi, X.: Distributions and light absorption property of water soluble organic carbon in a typical temperate glacier, southeastern Tibetan Plateau, Tellus B, 70, 1–15, https://doi.org/10.1080/16000889.2018.1468705, 2018.
- Wu, G., Ram, K., Fu, P., Wang, W., Zhang, Y., Liu, X., Stone, E. A., Pradhan, B. B., Dangol, P. M., Panday, A. K., Wan, X., Bai, Z., Kang, S., Zhang, Q., and Cong, Z.: Water-Soluble Brown Carbon in Atmospheric Aerosols from Godavari (Nepal), a Regional Representative of South Asia, Environ. Sci. Technol., 53,

**3471–3479, https://doi.org/10.1021/acs.est.9b00596, 2019.**

- 4. Page 4, Line 10-11: "34 samples of seasonal snow collected in December 2020 and January 2021", repeated information which has been mentioned in Line 20.
  R: The sentence has been revised as follows. (See Page 6, Line 15-18)
  "To address this deficiency, we make the primary investigation of the fluorescence characteristics, absorption properties, and radiative effects of WSOC in seasonal snow samples in northeastern China."
- 5. Page 5, Line 11: "sample sites" or "sampling sites"?R: Revised as "sampling sites" throughout the manuscript.
- 6. In Section 2.2, how did the authors quantify the concentration of WSOC in the snow samples? Did they randomly select different masses of snow samples for dissolution? Or snow need to be weighed? More information about experiment should be supplemented in the Experiment section.

R: All collected snow samples were stored in a freezer at -20 °C until analyzed in the laboratory. In the lab process, the samples were firstly melted at room temperature (25 °C). Then 30 mL meltwater was taken for each sample with the clean disposable syringe (Jiangnan, Anhui, China) and injected into the pre-baked (4h, 450 °C) bottle glass passing through 0.45 µm pore-sized polytetrafluoroethylene filters (Jinteng, Tianjin, China). Finally, the concentration of WSOC was measured by the total organic carbon analyzer (Aurora 1030W, OI Analytical, TX, USA). Additionally, a blank sample prepared with ultrapure water was also measured for blank correction before the sample measurement. No weighing of samples was required for the entire process.

The related content has been added to the manuscript. (See Page 9, Line 8-17)

7. Page 6, Line 27. It is recommended to provide how inner filtration effect, Raman scattering and Rayleigh scattering are removed in EEM.

R: The following content has been added. (See Page 10, Line 12-20)

"We normalized fluorescence intensity to that of the water Raman unit (RU), which exhibits a peak excitation wavelength of 350 nm, and deducted this Raman signal from all subsequent sample tests (Lawaetz and Stedmon, 2009). The inner filtration effect correction is based on the absorbance-based approach (Kothawala et al., 2013), using the measured absorbance at each pair of excitation and emission wavelengths across the EEMs to convert the observed fluorescence intensity into the corrected fluorescence intensity. Rayleigh scattering peaks were processed by interpolation algorithm in EEMscat MATLAB toolbox (Bahram et al. 2006)."

 Page 6, Line 29: "As fluorescence spectra with wavelengths greater than 600 nm are primarily noise", I would like to know why the authors measured excitation and emission wavelengths greater than 600 nm.

R: We noted that previous studies had selected the range of less than 600 nm in fluorescence spectrum measurements of WSOC in aquatic environment or aerosol (Chen et al., 2019; Zhao et al., 2017; Wu et al., 2021; Birdwell and Valsaraj, 2010). However, it could not be fully confirmed whether signals greater than 600 nm contain fluorescence information for snow samples. Therefore, we measured all the spectra from 240 nm to 800 nm, and found the signals larger than 600 nm are basically noise. We indicated that there is no obvious influence on our results after deduction, which is consistent with the study of Zhou et al. (2019).

9. Page 7, Line 28: Please check Eq. 3.

R: Corrected as suggested. (See Page 12, Line 6, Eq. 3)

$${}^{"}HIX = \frac{I(Ex = 254, Em = 435 - 480)}{I(Ex = 254, Em = 300 - 345) + I(Ex = 254, Em = 435 - 480)},$$
(3)

10. Please provide the absorption measurement of WSOC in the method section.
R: Provided as suggested. (See Page 12, Line 12-14) *"The absorption spectra of WSOC were derived from 240 to 800 nm in 3 nm*

intervals. The baseline shifts and scattering effects of the measurement for the absorption spectra were corrected by subtracting the average absorbance above 600 nm from the whole spectrum (Chen et al., 2019b)."

11. Please provide the blank concentration of WSOC in the method section.

R: Provided as suggested. (See Page 9, Line 15-17) "Additionally, a blank sample prepared with ultrapure water was measured for blank correction before the sample measurement. The blank concentration of WSOC is 0.35 mg L-1 and the blank-corrected WSOC concentrations are provided in Table S1."

- 12. Page 10, Line 1-6: Suggest put it in the above paragraph.R: We have put these sentences in the above paragraph as suggested. (See Page 15, Line 4-9)
- 13. Page 10, Line 23: The author should give the full name of "HULIS".
  R: Revised as suggested. (See Page 16, Line 17)
  "C1 exhibits a primary peak at Ex = 240 nm, Em = 448 nm, indicating a high-oxygenated humic-like substance (HULIS) found primarily in aromatic conjugated macromolecules (Chen et al., 2016)."
- 14. Page 10, Line 25: "Wen et al. 2021", format error.R: Revised as Wen et al. (2021).

15. Page 12, Line 15: Indeed, I don't know how the author obtained this conclusion, because the author only mentions SEIN and SNC here.
R: Sorry for the ambiguity. Revised as follows. (See Page 19, Line 4-9) *"Indeed, although HULIS-2 might be derived from any combination of terrestrial, anthropogenic, and microbial sources (Osburn et al., 2016; Stedmon et al., 2003; Yamashita et al., 2008; Zhang et al., 2009), human activity (e.g., agriculture, and microbial source)*

industrial emissions) is most intensive in SNC. Therefore, our combined analysis suggests that anthropogenic source is the main contributor to seasonal snow in SNC (Figs. 1a and 3d; Guo and Hu, 2022). The conclusion is also in good agreement with previous study (Zhou et al., 2019b)."

- 16. Page 14, Line 8: Can you really claim "monotonously"?R: We have revised "monotonously" as "generally". (See Page 21, Line 8)
- 17. Page 14, Line 14-16: "FI values of ≤1.4 correspond to terrestrial sources and values of ≥1.9 denote a primarily microbial origin, values of 1.4–1.9 suggest a mixed origin.". Suggest put it in Section 2. In addition, please provide references.

R: The sentence has been moved to Section 2 (See Page 12, Line 9-11) and the reference has been added as follows.

- McKnight, D. M., Boyer, E. W., Westerhoff, P. K., Doran, P. T., Kulbe, T., and Andersen, D. T.: Spectrofluorometric characterization of dissolved organic matter for indication of precursor organic material and aromaticity, Limnol. Oceanogr., 46, 38–48, https://doi.org/10.4319/lo.2001.46.1.0038, 2001.
- Page 15, Line 2: The result of TFV could not be seen in Figure 5a. Is it Figure 5b?
   as well as WSOC (280).

R: Sorry for the mistake. We have revised the analysis of Figure 5 to correspond with the figure (See Page 22, Line 9-12). The revised contents are as follows. *"Figure 5a describes the spatial distribution of*  $a_{WSOC}(280)$  *as WSOC absorption in the snowpack of northeastern China; Figure 5b depicts TFV as a measure of the spatial distribution of absolute WSOC fluorescence intensity for comparison."*

- 19. Missing space in front of the unit in Figure 5 and elsewhere.R: Revised.
- 20. Page 16, Line 19: "P" should be "P".

**R: Revised.**

21. Page 17, Line 1-11: Suggest putting it in Section 1 (Introduction) and revising it appropriately.

R: We have moved these sentences to Section 1 after appropriate revisions. (See Page 5, Line 20-22 to Page 6, Line 1-8)

"EEM-PARAFAC can only provide plausible information about the componentspecific influence of WSOC on fluorescent properties, and the quantitative fractional contributions of specific components to light absorption are still unknown. Recently, Chen et al. (2019a) collected atmospheric aerosol samples in PM2.5 over Xi'an, China, and successfully attributed the dithiothreitol (DTT) activity levels to various BrC components by coupling DDT and BrC datasets. A similar attribution method has been applied to various research areas, including climate change, extreme weather, and atmospheric environments (Cao et al., 2015; Pokrovsky, 2019; Xin et al., 2016; Zhao et al., 2019). In this study, we applied a multiple linear regression method comparable to that of Chen et al. (2019a) to derive the fractional contribution of each WSOC component to light absorption. Despite this method having been used elsewhere (Wu et al., 2022; Wu et al., 2021), it remains a highly innovative approach to evaluating the light absorption of snowpack WSOC."

22. Figure 7 should include the legend. In addition, in Line 11, Figure 7a refers to the absorption at wavelengths at 280-400 nm?R: Figure 7 has been revised as suggested and figure subtitles have been corrected

(See Page 28, Line 3). The revised figure is as follows.

Figure 7: Regional averages for the relative contributions of the three fluorescent components to light absorption at wavelengths of (a) 280–400 nm and (b) 280 nm.

23. Page 23, Line 5, this sentence belongs to Section 2.

R: Relevant contents in Section 2 have been modified as follows. (See Page 13, Line 16-18)

"As described in detail by Wang et al. (2017), the model involves parameters including solar zenith angle, impurity concentrations, mass absorption coefficient of impurities, and snow grain radius."

24. References: Please check the references and unified format. For example, "Characterization of Chromophoric Water-Soluble Organic Matter in Urban, Forest, and Marine Aerosols by HR-ToF-AMS Analysis and Excitation–Emission Matrix Spectroscopy," in Line 8.

R: We have been carefully revised and unified the references throughout the

manuscript.

**References:**

- Birdwell, J. E. and Valsaraj, K. T.: Characterization of dissolved organic matter in fogwater by excitation–emission matrix fluorescence spectroscopy, Atmos. Environ., 44, 3246–3253, https://doi.org/10.1016/j.atmosenv.2010.05.055, 2010.
- Chen, Q., Wang, M., Wang, Y., Zhang, L., Li, Y., and Han, Y.: Oxidative Potential of Water-Soluble Matter Associated with Chromophoric Substances in PM 2.5 over Xi'an, China, Environ. Sci. Technol., 53, 8574–8584, https://doi.org/10.1021/acs.est.9b01976, 2019.
- Wu, G., Fu, P., Ram, K., Song, J., Chen, Q., Kawamura, K., Wan, X., Kang, S., Wang, X., Laskin, A., and Cong, Z.: Fluorescence characteristics of water-soluble organic carbon in atmospheric aerosol, Environ. Pollut., 268, 115906, https://doi.org/10.1016/j.envpol.2020.115906, 2021.
- Zhou, Y., Wen, H., Liu, J., Pu, W., Chen, Q., and Wang, X.: The optical characteristics and sources of chromophoric dissolved organic matter (CDOM) in seasonal snow of northwestern China, The Cryosphere, 13, 157–175, https://doi.org/10.5194/tc-13-157-2019, 2019.
- Zhao, Y., Song, K., Shang, Y., Shao, T., Wen, Z., and Lv, L.: Characterization of CDOM of river waters in China using fluorescence excitation-emission matrix and regional integration techniques: DOC and CDOM in Rivers in China, J. Geophys. Res.-Biogeosci., 122, 1940–1953, https://doi.org/10.1002/2017JG003820, 2017.

---

## Editor Decision (ED1)

**Additional comments to authors-acp-2022-336**

Based on the comments of two reviewers, who support the publication, and after my consideration, I have decided that the manuscript is of suitable atmospheric interest to merit publication in *Atmospheric Chemistry and Physics*. The results of this manuscript present useful information on the important impact of water-soluble organic carbon (WSOC) on snow albedo and the solar radiation balance. The authors have thoroughly responded all the questions/comments raised by the reviewers, and modified the manuscript according to the suggestions.

However, I have still some additional comments, which are needed to be solved before publication.

**Comments/ errors:** (Lines as in the revised version of MS with tracked changes)

Line 13/page 4: "And" can be deleted.

Line 16/17, page 6: Please correct as: »To address this deficiency, we made the first investigation…«

Line 9, page 9: Correct: "… until analysis." "… were first melted" (not firstly)
Lines 15-17, page 9: Please, correct (awkwardly written).

Line 16, page 14: Number of significant digits are now mostly good; but …"18.0 µg g$^{-1}$" should also be changed (18 µg g$^{-1}$). Please, check again your results throughout the manuscript!

Line 7, page 34: "….of WSOC, indicating that snowpack WSOC originates…" It should be "..of WSOC indicate that…".

Reviewer 2: 13. Page 10, Line 23: The author should give the full name of "HULIS".
No need, HULIS is already mentioned in the abstract. Just name it appropriately (humic-like substances).

Please, check again the English language; there are still errors.

---

## Author Response (AR2)

**Response to additional comments to authors-acp-2022-336**

Based on the comments of two reviewers, who support the publication, and after my consideration, I have decided that the manuscript is of suitable atmospheric interest to merit publication in Atmospheric Chemistry and Physics. The results of this manuscript present useful information on the important impact of water-soluble organic carbon (WSOC) on snow albedo and the solar radiation balance. The authors have thoroughly responded all the questions/comments raised by the reviewers, and modified the manuscript according to the suggestions.

However, I have still some additional comments, which are needed to be solved before publication.

R: Thanks for the editor's careful reading, helpful comments, and constructive suggestions, which have significantly improved the presentation of our manuscript. We have carefully revised our manuscript, and the related typos and grammar errors have been corrected accordingly.

Comments/ errors: (Lines as in the revised version of MS with tracked changes)

1.  Line 13/page 4: "And" can be deleted.

    R: Revised as suggested. (Line 20, page 4)

2.  Line 16/17, page 6: Please correct as: »To address this deficiency, we made the first investigation…«

    R: Corrected as suggested. (Line 1-2, page 7)

    *"To address this deficiency, we made the first investigation of the fluorescence characteristics, absorption properties, and radiative effects of WSOC in seasonal snow samples in northeastern China."*

3.  Line 9, page 9: Correct: "... until analysis." "... were first melted" (not firstly).

    R: Corrected as suggested. (Line 9, page 9)

    *"All collected snow samples were stored in a freezer at −20 °C until analysis in the*

*laboratory. In the lab process, the samples were first melted at room temperature (25 ℃).*”

4. Lines 15-17, page 9: Please, correct (awkwardly written).

   R: Corrected as suggested. (Lines 17-19, page 9)

   *“The concentration of WSOC for ultrapure water blank is 0.35 mg L$^{-1}$, and the value of each sample after blank subtraction is presented in Table S1.”*

5. Line 16, page 14: Number of significant digits are now mostly good; but ...”18.0 μg g$^{-1}$” should also be changed (18 μg g$^{-1}$). Please, check again your results throughout the manuscript!

   R: Revised as suggested throughout the manuscript. (Line 16, page 14).

6. Line 7, page 34: “….of WSOC, indicating that snowpack WSOC originates…” It should be “..of WSOC indicate that…”.

   R: Revised as suggested. (Line 7, page 34)

   *“The high degree of humification and minimal bioavailability of WSOC indicate that snowpack WSOC originates primarily from soil sources.”*

7. Reviewer 2: 13. Page 10, Line 23: The author should give the full name of “HULIS”. No need, HULIS is already mentioned in the abstract. Just name it appropriately (humic-like substances).

   R: Revised as suggested. (Line 19-20, Page 1; Line 20, Page 16)

8. Please, check again the English language; there are still errors.

   R: We have polished the manuscript very carefully.